# Associations between Subsequent Hospitalizations and Primary Ambulatory Services Utilization within the First Year after Acute Myocardial Infarction and Long-Term Mortality

**DOI:** 10.3390/jcm9082528

**Published:** 2020-08-05

**Authors:** Ygal Plakht, Dan Greenberg, Harel Gilutz, Jonathan Eli Arbelle, Arthur Shiyovich

**Affiliations:** 1Department of Nursing, Faculty of Health Sciences, Ben-Gurion University of the Negev, Beer-Sheva 8410501, Israel; 2Department of Emergency Medicine, Soroka University Medical Center, Beer-Sheva 8410501, Israel; 3Department of Health Systems Management, School of Public Health, Faculty of Health Sciences, Ben-Gurion University of the Negev, Beer-Sheva 8410501, Israel; dangr@bgu.ac.il; 4Goldman Medical School, Faculty of Health Sciences, Ben-Gurion University of the Negev, Beer-Sheva 8410501, Israel; Gilutz@bgu.ac.il (H.G.); arbelle_J@mac.org.il (J.E.A.); 5Southern District, Maccabi Healthcare Services, Beer-Sheva 8410501, Israel; 6Department of Cardiology, Rabin Medical Center, Petah Tikva 49100, Israel; arthur.shiyovich@gmail.com; 7Sackler Faculty of Medicine, Tel Aviv University, Tel Aviv 6997801, Israel

**Keywords:** acute myocardial infarction, healthcare resource utilization, mortality, prognosis

## Abstract

Healthcare resource utilization peaks throughout the first year following acute myocardial infarction (AMI). Data linking the former and outcomes are sparse. We evaluated the associations between subsequent length of in-hospital stay (SLOS) and primary ambulatory visits (PAV) within the first year after AMI and long-term mortality. This retrospective analysis included patients who were discharged following an AMI. Study groups: low (0–1 days), intermediate (2–7) and high (≥8 days) SLOS; low (<10) and high (≥10 visits) PAV, throughout the first post-AMI year. All-cause mortality was set as the primary outcome. Overall, 8112 patients were included: 55.2%, 23.4% and 21.4% in low, intermediate and high SLOS groups respectively; 26.0% and 74.0% in low and high-PAV groups. Throughout the follow-up period (up to 18 years), 49.6% patients died. Multivariable analysis showed that an increased SLOS (Hazard ratio (HR) = 1.313 and HR = 1.714 for intermediate and high vs. low groups respectively) and a reduced number of PAV (HR = 1.24 for low vs. high groups) were independently associated with an increased risk for mortality (*p* < 0.001 for each). Long-term mortality following AMI is associated with high hospital and low primary ambulatory services utilization throughout the first-year post-discharge. Measures focusing on patients with increased SLOS and reduced PAV should be considered to improve patient outcomes.

## 1. Introduction

Throughout the past few decades, dramatic changes have been introduced to most aspects of acute myocardial infarction (AMI) management, resulting in significant improvement of patient outcomes [1,2,3,4,5]. However, AMI survivors continue to be at excessive risk of long-term morbidity and mortality as well as of increased healthcare resource utilization and economic burden, particularly in the first year following the index event [6,7,8,9]. Risk stratification of AMI survivors could improve long-term outcomes and target allocation of resources. Identification of risk factors for long-term outcomes after AMI have focused mainly on data available at the time of the initial hospitalization for AMI [10,11]. Wang et al. [12] recently showed that in-hospital admissions after AMI were associated with the risk of a subsequent AMI.

Data evaluating the associations between healthcare resource utilization following AMI and mortality are scant. The aim of the current study was to evaluate the association between length of stay of subsequent hospitalizations (SLOS) and the number of primary ambulatory clinic visits (PAV) during the first post-AMI year and long-term mortality.

## 2. Materials and Methods

### 2.1. Study Population

This retrospective observational study evaluated patients who were admitted with and survived an AMI from 1.1.2002 through 12.31.2012 to a tertiary medical center (Soroka University Medical Center) in Southern Israel. Patients were excluded due to the following criteria: (1) death during the index AMI admission or during the first year after discharge; (2) not citizens of the State of Israel, (3) insured by neither Maccabi Healthcare Services nor Clalit Health Services (the two largest health plans in Israel) and (4) recurrent hospitalizations for AMI during the study period (only the first AMI admission (index admission) was included).

The local ethics committees approved the study (Soroka University Medical Center, Maccabi Healthcare Services and Clalit Health Services, identifiers; SOR-0167-12, 28/2014, COM-0071-13, respectively), which was performed in accordance with the Helsinki declaration. Exemption from informed consent was granted and personal details of investigated patients remained confidential.

### 2.2. Data Sources and Classifications

We obtained the data from the electronic medical records of the Soroka University Medical Center, Maccabi Healthcare Services and Clalit Health Services. Mortality data were obtained from the Ministry of the Interior population registry. Individual patient-level data from the different databases were linked using the unique personal identification number, followed by coding into a study identification number prior to further data processing.

The baseline data included demographics and clinical characteristics of the index hospitalizations (comorbidities, type of AMI, work-up as well as interventions administered for the AMI), as previously reported for the Soroka Acute Myocardial Infarction (SAMI) project [10,13].

The diagnosis of AMI was based on the international Classification of Diseases, Ninth Revision, Clinical Modification (ICD-9-CM) codes: ST-elevation AMI (STEMI) 410.0*–410.6* and Non-ST-elevation AMI (NSTEMI) 410.7*–410.9*. Grouping of diseases and interventions were based on ICD-9-CM discharge codes [10,13].

Adherence with the following guideline-recommended medical therapy was evaluated: Aspirin, Statins, Beta blockers, Angiotensin-converting enzyme (ACE) inhibitors or Angiotensin II receptor blockers (ARBs). The adherence was calculated as the rate of issued monthly prescriptions throughout the first year following hospital discharge. Patients with an adherence of 80% or more for all the evaluated medication groups were classified as adherent while the rest were classified as non-adherent [14,15,16]. Based on previous reports, a one-year adherence interval (as applied herein) reflects the adherence plateau, beyond which adherence leveled off [17,18].

### 2.3. Healthcare Services Utilization and the Study Groups

Data regarding hospitalizations and primary ambulatory services utilization during the first year after discharge were obtained from the databases of the two health plans [19,20]. All-cause subsequent hospitalizations and their length (SLOS) and number of PAV were quantified. The SLOS groups were defined as: low hospital utilization (0–1 days); intermediate utilization (2–7 days) and high utilization (>8 days). In addition, the number of PAV were divided into four equally sized groups (quartiles) followed by their merging into two groups according to the strength of the univariate association with the dependent variable. Thus, two groups of primary ambulatory healthcare services utilization were created in accordance with the number of PAV: <10 visits (low utilization) and ≥10 visits (high utilization).

### 2.4. Propensity Scores

We calculated propensity scores or predicted probabilities of the healthcare services utilization (SLOS and PAV) based on a set of relevant patient characteristics using a two-step procedure: (1) the parameters (SLOS and PAV) were modeled as the outcome variables in a general linear model included as predictor variables a selected covariates that were identified as key confounders and (2) predicted probabilities of SLOS and PAV (“propensity scores”) were used as covariates in the final regression model (propensity score adjustment).

A total of 34 covariates (including age, sex, nationality, cardiovascular risk factors, other comorbidities, type of AMI, type of treatment and compliance with the medications) were initially included in the propensity score model. Of those, 23 variables consisted of the propensity scores.

### 2.5. Follow-Up and Outcomes

Follow-up started from the second year (≥day 366 following the date of the discharge from the index hospitalization) and continued up to 18 years (or till 7.1.2020). The primary outcome was all-cause mortality.

### 2.6. Statistical Analysis

Statistical analysis was performed using IBM SPSS 26 (SPSS Inc., Chicago, IL, USA) Statistics software. Patient characteristics were presented as mean and standard deviation (SD) for continuous variables and numbers (*n*) and percent (%) for the categorical data. In addition, SLOS and PAV parameters were presented as median and inter-quartile range (IQR). The comparison of baseline characteristics between the study groups was performed using Chi-square test/Chi-square test for linear trend for categorial variables and Student’s t-test/analysis of variance (ANOVA) for linear trend for continuous variables for PAV and SLOS categories respectively. The comparison of outcomes between the study groups was performed using the survival approach. The univariate analysis compared the risk of mortality with the creation of survival functions (Kaplan–Meier) using the Log-rank test. In addition, we used Cox regression analysis to estimate the relative risk of long-term mortality for the study groups. Four models were built: the first and second models were the univariate models which included the variables of SLOS and PAV (a separate model for each). The third model included the variables of SLOS and PAV together. Finally, the forth model included these above variables and the investigated baseline characteristics which were statistically related to the outcome and the propensity scores. The results of the models were presented as the regression coefficients (B) and their standard errors (SE), hazard ratios (HR) and 95% Confidence intervals (CI) for HR. For each test, *p* < 0.05 was considered as statistically significant.

## 3. Results

### 3.1. Study Population and Groups

Overall, 8112 of 12,503 post-AMI patients were included in the current study. The reasons patients were excluded were as follows: death during the index AMI admission or during the first year after discharge (*n* = 1931), not citizens of the State of Israel (*n* = 414) and not insured by Maccabi Healthcare Services or Clalit Health Services (*n* = 2046). The mean SLOS was 5.98 (SD = 15.5) days, median 1 day (IQR: 0–6); 4068 (50.2%) of patients were admitted to hospital for one day or longer throughout the first year after the index hospitalization.

The mean SLOS of patients hospitalized for one day or more throughout the first post-AMI year was 11.9 (SD = 20.3) days, a median of 6 days (IQR: 3–13). There were 4474 (55.2%) patients included in the low hospital services utilization group; 1899 (23.4%) in the intermediate hospital services utilization group; and 1739 (21.4%) in the high hospital services utilization group.

A total of 7938 (97.9%) patients has at least one PAV throughout the first follow-up year, with a mean of 18.7 (SD = 12.2), a median of 17 (IQR: 10–25) visits. The distribution according to the number of PAV was as following: 2111 (26%) patients were included in the group of low ambulatory services utilization and the rest, consisting of 6001 (74%) patients, comprised the high ambulatory utilization group. A negative significant association between the SLOS and the number of PAV was found (*p* < 0.001).

Table 1 displays patient characteristics according to the groups of SLOS and the number of PAV throughout the first year following AMI. Greater SLOS was associated with increased patient age, while greater number of PAV was related to a mildly reduced age. Male patients had lower SLOS with no difference between the sexes in PAV. The minorities (Arabs) were characterized by lower PAV compared with Jews, with no significant differences in SLOS between these groups. Higher utilization of both services was associated with an increased prevalence of cardiovascular risk factors, except smoking and family history of cardiovascular diseases. Congestive heart failure as well as most non-cardiovascular comorbidities were also more prevalent among patients with greater SLOS and PAV. Presentation as STEMI tended to be more prevalent among patients with higher SLOS and PAV (borderline significance for the latter). In-hospital stay of 7 days or more at the index admission was associated with greater SLOS and PAV. Patients who underwent percutaneous coronary intervention (PCI) and those who underwent coronary artery bypass graft surgery (CABG) had reduced SLOS but greater PAV. Higher adherence to guideline-recommended medical therapy throughout the first year was observed in patients with lower SLOS and a greater number of PAV (*p* < 0.001 for each, see also Appendix A).

Abbreviates: SLOS, Length of stay of subsequent hospitalizations; PAV, Primary ambulatory visits; SD, standard deviation; CIHD, Chronic ischemic heart disease; PVD, Peripheral vascular disease; IHD, Ischemic heart disease; COPD, Chronic obstructive pulmonary disease; AMI, Acute myocardial infarction; STEMI, ST segment elevation myocardial infarction; ICCU, Intensive cardiac care unit; LOS, Length of stay; PCI, Percutaneous coronary intervention; CABG, Coronary artery bypass surgery; LV, Left ventricular.

### 3.2. Follow-Up and Outcomes

The follow-up lasted from 366 up to 6575 days (18 years) post-hospital discharge with a median follow-up of 3334 days (9.1 years). During the follow-up period, 4021 (49.6%) patients died with a cumulative mortality of 0.63. Mortality data and survival curves according to healthcare services utilization are presented in Table 1 and Figure 1. Higher SLOS was significantly associated with increased mortality while the opposite association was found with PAV (*p* < 0.001 for each).

### 3.3. Univariate Analysis

The results of the univariate models (Table 2, models a and b) showed an increased risk of approximately 1.7 and 3.0 for long-term mortality in the groups of intermediate and the high hospital healthcare services utilization group respectively as compared with the group of low utilization. In contrast, high primary ambulatory services utilization was associated with approximately 1.2 decreased risk of mortality.

### 3.4. Multivariable Analysis

The results of the univariate analysis were consistent with the findings of the multivariable models before (Table 2, model c) and after (Table 2, model d) adjustment for potential confounders. The results of the multivariable models have shown that increased SLOS (HR ~ 1.3. and 1.7 for intermediate and high as compared with low SLOS group) and reduced number of PAV (HR ~ 1.2 for low as compared with high PAV group) were significantly associated with increased risk of mortality (*p* < 0.001 for each). Most of the investigated comorbidities (age, heart diseases, cardiovascular and non-cardiovascular risk factors) were associated with an increased risk of long-term mortality (except for “family history of ischemic heart disease” which had a negative association with the outcome). Additionally, invasive treatment of AMI (PCI and CABG) seems to be related with a lower risk of dying.

## 4. Discussion

In the current study, from a large Israeli hospital combined with ambulatory data, we evaluated the association between the extent of healthcare resource utilization during the first year following AMI and long-term survival. The main findings include a significant, independent, dose-response like association between the extent of hospital healthcare services utilization and long-term all-cause mortality. Furthermore, a significant reverse association was found between primary ambulatory services utilization throughout first year and long-term mortality.

Various studies have consistently shown that healthcare services utilization are highest throughout the first year following an AMI [7,8,9,21,22]. The association between hospital readmission following AMI and worse subsequent outcomes has been previously reported and is consistent with our findings regarding SLOS and outcomes [23,24,25]. However, our study adds to the current knowledge by (1) evaluating SLOS, though obviously associated with readmissions, focusing on the total time spent in the hospital rather than just the number of readmissions and (2) focusing on a long period of time (a year) following AMI and a relatively long follow-up period for the outcome.

Several potential mechanisms could explain our findings. It is plausible that increased SLOS, as previously reported for readmissions, results at least partially from increased co-morbidity as well as worse outcomes following the index AMI (i.e., reduced left ventricular function) [26,27]. Although we adjusted for many of these confounders, it is still possible that additional unaccounted confounders or an underappreciation of the severity of some of the existing confounders could explain our findings. An additional explanation for our findings could be the post-discharge management of these AMI patients. Better post-discharge management and better compliance are associated with improved patient outcomes [15,16,17,28,29,30,31].

We believe that our finding of a reverse association between ambulatory visits and mortality strongly supports the latter explanation since low (especially very low) numbers of such ambulatory visits probably represents, to some extent, undertreatment and reduced compliance. Furthermore, we have actually shown the latter associated in the current study: increased compliance with guideline-recommended medical therapy throughout the first year was associated with a higher number of primary care visits and reduced SLOS. Furthermore, the common reasons for subsequent hospitalizations throughout the first year following AMI (e.g., diabetes mellitus, anemia, heart failure, pneumonia, gastrointestinal hemorrhage, renal failure, and complications of an implant or graft) have been shown to be associated with increased risk of major cardiovascular events [12,32]. Moreover, hospitalizations themselves could actually be a risk factor for negative outcomes and mortality, mediated by an increased risk of infections, stress, inflammation and depression [12].

Interestingly, we found that male patients had lower SLOS compared with females, yet the PAVs were similar between the sexes. Is seems that the most plausible reason for these findings is the significant age difference between the sexes (with females approximately eight years older than male patients) as a strong positive association between age and SLOS (but not PAV) was observed. Nevertheless, sex-related disparities in other characteristics and in in-hospital and post-discharge management of AMI patients were previously described and could, at least partially, explain reduced hospital referral and admission (both self-referral and referral by caregivers [33,34].

### Limitations

Several limitations of the current study should be mentioned. First, this is a single center (single country) retrospective observational study which shares the limitations of such a design and could have limited external validity. Second, we did not differentiate cardiovascular versus non-cardiovascular-related healthcare services utilization and causes of mortality. Third, we used administrative data which may be subject to recording bias. Fourth, private visits or even hospitalizations which were paid by patients out of their own pocket were not accounted for. Fifth, we did not evaluate the potential causality between SLOS and mortality. Sixth, evaluating long-term outcomes, we excluded patients who died throughout the first year, hence the observed associations are not applicable to this subgroup. Furthermore, the choice of one year for SLOS and PAV, although based on previous reports of peaks in healthcare services utilization, is somewhat arbitrary and might not necessarily represent the optimal predictive period following admission. Seventh, the current study did not include citizens from relatively small insurers (< 20%) and patients who were not Israeli citizens. Although this could be a potential source of bias, all insurers (plans) in Israel must by law accept citizens regardless of any preexisting conditions or health status and provide relatively similar coverage regarding the investigated services, hence this is unlikely to significantly bias our findings. Eighth, adherence rates were calculated based on computerized dispensing records which might not fully represent actual medication taking. However, using computerized records diminished the possibility of recall or self-reporting bias. Finally, we did not collect information regarding the rate of administration and adherence with dual anti-platelet therapy.

## 5. Conclusions and Clinical Implications

The current study demonstrated a potential, “dose-response like” association between SLOS throughout one-year following AMI and long-term all-cause mortality. Furthermore, a reverse association was found between the number of PAV throughout the first year and long-term mortality. The findings of this study suggest that clinicians should focus on patients with subsequent admissions and those with reduced primary care visits following AMI, in order to identify the patient-specific reasons for these negative prognostic factors, followed by custom-tailored interventions (e.g., increase PAVs and secondary prevention measures) to improve patient outcomes. Furthermore, our findings can assist decision makers and healthcare providers in long-term risk stratification of AMI survivors. Moreover, continuity of care and optimal transfer of medical information between the primary care facilities and hospitals are important to enable recognition of targets and interventions to improve long-term outcomes.

## Figures and Tables

**Figure 1 jcm-09-02528-f001:**
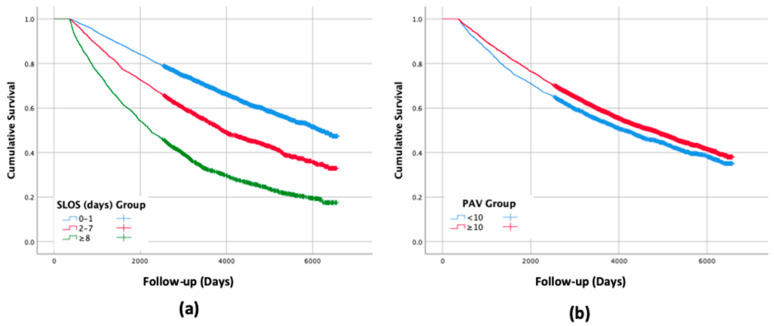
Survival curves for long-term mortality by the groups of: (**a**) Length of stay of subsequent hospitalizations (SLOS); (**b**) Number of and primary ambulatory visits (PAV) throughout the first year following acute myocardial infarction (AMI) (*p* < 0.001 for each). Abbreviates: SLOS—Length of stay of subsequent hospitalizations, PAV—Primary ambulatory visits.

**Table 1 jcm-09-02528-t001:** Baseline characteristics and mortality data of the patients by the groups of length of stay of subsequent hospitalizations (SLOS) and number of primary ambulatory visits (PAV) throughout the first year following acute myocardial infarction (AMI).

Parameter	Total	SLOS (Days)			*p* for Trend	PAV (Number)		*p*
		0–1	2–7	≥8		<10	≥10	
*n*	8112	4474	1899	1739		2111	6001	
***Demographics***								
**Age (years) *,** Mean (SD)	64.97 (13.58)	62.76 (13.57)	66.15 (13.64)	69.37 (12.22)	<0.001	65.53 (14.71)	64.77 (13.15)	0.038
<65	4098 (50.5)	2608 (58.3)	882 (46.4)	608 (35.0)	<0.001	1056 (50)	3042 (50.7)	0.002
65–75	1964 (24.2)	951 (21.3)	484 (25.5)	529 (30.4)		432 (20.5)	1532 (25.5)	
≥75	2050 (25.3)	915 (20.5)	533 (28.1)	602 (34.6)		623 (29.5)	1427 (23.8)	
Sex (males) *	5663 (69.8)	3362 (75.1)	1290 (67.9)	1011 (58.1)	<0.001	1470 (69.6)	4193 (69.9)	0.839
Minorities *	1839 (22.7)	1026 (22.9)	438 (23.1)	375 (21.6)	0.309	550 (26.1)	1289 (21.5)	<0.001
***Cardiac diseases***								
Cardiomegaly	609 (7.5)	282 (6.3)	143 (7.5)	184 (10.6)	<0.001	150 (7.1)	459 (7.6)	0.415
Supraventricular arrhythmias *	1191 (14.7)	470 (10.5)	304 (16.0)	417 (24.0)	<0.001	239 (11.3)	952 (15.9)	<0.001
Congestive heart failure *	1161 (14.3)	431 (9.6)	281 (14.8)	449 (25.8)	<0.001	301 (14.3)	860 (14.3)	0.935
Pulmonary heart disease	599 (7.4)	236 (5.3)	161 (8.5)	202 (11.6)	<0.001	143 (6.8)	456 (7.6)	0.213
CIHD *	6482 (79.9)	3675 (82.1)	1501 (79.0)	1306 (75.1)	<0.001	1558 (73.8)	4924 (82.1)	<0.001
Atrioventricular block	297 (3.7)	153 (3.4)	57 (3.0)	87 (5.0)	0.012	59 (2.8)	238 (4.0)	0.014
***Cardiovascular risk factors***								
Renal diseases *	2706 (33.4)	1180 (26.4)	663 (34.9)	863 (49.6)	<0.001	660 (31.3)	2046 (34.1)	0.018
Diabetes Mellitus *	3014 (37.2)	1429 (31.9)	738 (38.9)	847 (48.7)	<0.001	671 (31.8)	2343 (39.0)	<0.001
Dyslipidemia *	5596 (69.0)	3173 (70.9)	1323 (69.7)	1100 (63.3)	<0.001	1336 (63.3)	4260 (71.0)	<0.001
Hypertension *	4230 (52.1)	2192 (49.0)	1041 (54.8)	997 (57.3)	<0.001	1032 (48.9)	3198 (53.3)	<0.001
Obesity *	1861 (22.9)	1053 (23.5)	412 (21.7)	396 (22.8)	0.272	439 (20.8)	1422 (23.7)	0.006
Smoking *	3328 (41.0)	2047 (45.8)	757 (39.9)	524 (30.1)	<0.001	906 (42.9)	2422 (40.4)	0.040
PVD *	911 (11.2)	390 (8.7)	231 (12.2)	290 (16.7)	<0.001	219 (10.4)	692 (11.5)	0.148
Family history of IHD	594 (7.3)	428 (9.6)	110 (5.8)	56 (3.2)	<0.001	143 (6.8)	451 (7.5)	0.261
***Other disorders***								
COPD *	553 (6.8)	212 (4.7)	152 (8.0)	189 (10.9)	<0.001	137 (6.5)	416 (6.9)	0.488
Neurological disorders *	1174 (14.5)	497 (11.1)	302 (15.9)	375 (21.6)	<0.001	394 (18.7)	780 (13.0)	<0.001
Malignancy	239 (2.9)	105 (2.3)	55 (2.9)	79 (4.5)	<0.001	43 (2.0)	196 (3.3)	0.004
Anemia *	4225 (52.1)	2031 (45.4)	1026 (54.0)	1168 (67.2)	<0.001	988 (46.8)	3237 (53.9)	<0.001
Schizophrenia/Psychosis	120 (1.5)	57 (1.3)	26 (1.4)	37 (2.1)	0.02	50 (2.4)	70 (1.2)	<0.001
Gastro-intestinal bleeding	160 (2.0)	72 (1.6)	38 (2.0)	50 (2.9)	0.002	42 (2.0)	118 (2.0)	0.947
Alcohol/drug addiction	153 (1.9)	81 (1.8)	34 (1.8)	38 (2.2)	0.386	51 (2.4)	102 (1.7)	0.037
***Characteristics of the index hospitalization***								
Type of AMI, STEMI *	4091 (50.4)	2478 (55.4)	891 (46.9)	722 (41.5)	<0.001	1103 (52.3)	2988 (49.8)	0.052
Admitted/transposed to ICCU*	5727 (70.6)	3446 (77.0)	1259 (66.3)	1022 (58.8)	<0.001	1373 (65.0)	4354 (72.6)	<0.001
**LOS (days),*** Mean (SD)	10.16 (8.37)	9.49 (7.43)	9.80 (6.82)	12.29 (11.32)	<0.001	9.72 (8.51)	10.32 (8.32)	0.004
>7	3961 (48.8)	1968 (44.0)	952 (50.1)	1041 (59.9)	<0.001	909 (43.1)	3052 (50.9)	<0.001
**Type of treatment,** Noninvasive	1985 (24.5)	849 (19.0)	518 (27.3)	618 (35.5)	<0.001	690 (32.7)	1295 (21.6)	<0.001
PCI *	4993 (61.6)	2969 (66.4)	1122 (59.1)	902 (51.9)		1236 (58.6)	3757 (62.6)	
CABG *	1133 (14.0)	655 (14.6)	259 (13.6)	219 (12.6)		185 (8.8)	948 (15.8)	
***Results of echocardiography***								
*Echocardiography performance*	6692 (82.5)	3847 (86.0)	1528 (80.5)	1317 (75.7)	<0.001	1653 (78.3)	5039 (84.0)	<0.001
Severe LV dysfunction ^1,^*	627 (9.4)	257 (6.7)	149 (9.8)	221 (16.8)	<0.001	156 (9.4)	471 (9.3)	0.913
LV hypertrophy ^1^	282 (4.2)	133 (3.5)	68 (4.5)	81 (6.2)	<0.001	71 (4.3)	211 (4.2)	0.850
Mitral regurgitation ^1^	351 (5.2)	120 (3.1)	96 (6.3)	135 (10.3)	<0.001	74 (4.5)	277 (5.5)	0.106
Tricuspidal regurgitation ^1^	212 (3.2)	81 (2.1)	35 (2.3)	96 (7.3)	<0.001	51 (3.1)	161 (3.2)	0.825
Pulmonary hypertension ^1^	432 (6.5)	173 (4.5)	103 (6.7)	156 (11.8)	<0.001	106 (6.4)	326 (6.5)	0.935
**Compliance to the medical treatment ^2,^***	1767 (21.8)	1067 (23.8)	406 (21.4)	294 (16.9)	<0.001	353 (16.7)	1414 (23.6)	<0.001
**Long-term mortality**								
Deaths	4021 (49.6)	1691 (37.8)	1037 (54.6)	1293 (74.4)	<0.001	1133 (53.7)	2888 (48.1)	<0.001
Cumulative mortality	0.627	0.526	0.671	0.825	<0.001	0.649	0.62	<0.001

The data are presented as *n* (%) unless otherwise stated. * The parameters that were included into propensity scores. ^1^ Among persons with the results of echocardiography. ^2^ Compliance to the medical treatment relates to guideline recommended medical therapy during the first year after discharge from the index hospitalization.

**Table 2 jcm-09-02528-t002:** Relationships between length of stay of subsequent hospitalizations (SLOS) and number of and primary ambulatory visits (PAV) throughout the first year following acute myocardial infarction (AMI) and the risk of long-term all-cause mortality: (**a**) this model included only SLOS variables; (**b**) this model included only PAV variables; (**c**) this model included both SLOS and PAV variables; and (**d**) this model included both SLOS and PAV variables adjusted for the investigated baseline characteristics and propensity scores.

Model	Parameter	B (SE)	HR	(95% CI)	*p*
a.	SLOS (days), 0–1		1 (ref.)		
	2–7	0.504 (0.039)	1.656	(1.533; 1.789)	<0.001
	≥8	1.082 (0.037)	2.952	(2.745; 3.174)	<0.001
b.	PAV (number), <10		1 (ref.)		
	≥10	−0.148 (0.035)	0.863	(0.806; 0.924)	<0.001
c.	SLOS (days), 0–1		1 (ref.)		
	2–7	0.565 (0.040)	1.759	(1.627; 1.903)	<0.001
	≥8	1.160 (0.038)	3.190	(2.961; 3.437)	<0.001
	PAV (number), <10		1 (ref.)		
	≥10	−0.353 (0.036)	0.694	(0.647; 0.745)	<0.001
d.	SLOS (days), 0–1		1 (ref.)		
	2–7	0.273 (0.040)	1.313	(1.214; 1.421)	<0.001
	≥8	0.539 (0.039)	1.714	(1.587; 1.852)	<0.001
	PAV (number), <10		1 (ref.)		
	≥10	−0.211 (0.037)	0.809	(0.753; 0.870)	<0.001
	Age (years), <65		1 (ref.)		
	65–75	0.960 (0.045)	2.612	(2.392; 2.852)	<0.001
	≥75	1.538 (0.047)	4.653	(4.243; 5.103)	<0.001
	Minorities	−0.138 (0.043)	0.871	(0.800; 0.948)	0.001
	Supraventricular arrhythmias	0.360 (0.065)	1.434	(1.261; 1.630)	<0.001
	Congestive heart failure	0.154 (0.047)	1.167	(1.065; 1.279)	0.001
	Renal diseases	0.244 (0.039)	1.277	(1.183; 1.378)	<0.001
	Diabetes Mellitus	0.468 (0.040)	1.597	(1.476; 1.728)	<0.001
	Peripheral vascular disease	0.237 (0.046)	1.268	(1.159; 1.387)	<0.001
	Family history of IHD	−0.606 (0.122)	0.545	(0.429; 0.692)	<0.001
	COPD	0.472 (0.069)	1.603	(1.400; 1.836)	<0.001
	Malignancy	0.375 (0.086)	1.454	(1.229; 1.721)	<0.001
	Anemia	0.236 (0.044)	1.266	(1.161; 1.380)	<0.001
	Severe LV dysfunction	0.261 (0.058)	1.298	(1.158; 1.456)	<0.001
	LV hypertrophy	0.353 (0.075)	1.423	(1.229; 1.648)	<0.001
	Pulmonary hypertension	0.265 (0.059)	1.304	(1.161; 1.464)	<0.001
	Type of treatment, Noninvasive		1 (ref.)		
	PCI	−0.356 (0.045)	0.701	(0.642; 0.765)	<0.001
	CABG	−0.439 (0.071)	0.645	(0.561; 0.740)	<0.001
	Propensity score for SLOS	0.058 (0.007)	1.059	(1.045; 1.074)	<0.001
	Propensity score for PAV	−0.062 (0.011)	0.940	(0.920; 0.960)	0.002

Abbreviates: B, regression coefficient; SE, Standard error; HR, Hazard ratio; CI, Confidence interval; SLOS, Length of stay of subsequent hospitalizations; Ref., Reference group; PAV, Primary ambulatory visits; IHD, Ischemic heart disease; COPD, Chronic obstructive pulmonary disease; LV, Left ventricular; PCI, Percutaneous coronary intervention; CABG, Coronary artery bypass surgery.

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
