# Peer review of "Associations between Subsequent Hospitalizations and Primary Ambulatory Services Utilization within the First Year after Acute Myocardial Infarction and Long-Term Mortality"

_jcm, 2020, doi:10.3390/jcm9082528_

Round 1

Reviewer 1 Report

This retrospective observational study evaluated patients who were admitted with and survived an AMI from January 1,2002 through December 31, 2012 in a single tertiary medical center in Israel. The aim of this study was to evaluate the association between long-term mortality and the length of stay of subsequent hospitalizations (SLOS), as well as the number of primary ambulatory clinic visits (PAV), during the first post-AMI year. The authors showed that long-term mortality following AMI is associated with high SLOS and low primary ambulatory (PAV) services utilization during the first-year post-discharge. The conclusion of the study was that measures focusing on patients with increased SLOS and reduced PAV should be considered to improve patient outcomes.

This study has several serious limitations and raises multiple questions:

  1. Why were the patients who died either during the index AMI admission or during the first year after discharge excluded from the analysis?
  2. Why were patients who were not citizens of the state of Israel excluded?
  3. Why were patients who were not insured by Maccabi Healthcare Services or Clalit Health Services (the two largest health plans in Israel) excluded? What about other patients who were insured by other companies? Did they have not have insurance? If not, they would seem to be a more vulnerable group of patients and their inclusion might have changed the results of the study.
  4. What about patients who didn’t sign consent forms? Were they also included in this study?
  5. The authors mentioned that in this study they calculated propensity score or predicted probabilities of the healthcare services utilization (SLOS and PAV) based on a set of relevant patient characteristics using a two-step procedure: 1) the parameters (SLOS and PAV) were modeled as the outcome variables in a general linear model included as predictor variables and selected covariates identified as  key confounders, and 2) predicted probabilities of SLOS and PAV (“propensity scores”) were used  as covariates in the final regression model (propensity score adjustment).  A total of 34 covariates (including age, sex, nationality, cardiovascular risk factors, other comorbidities, type of AMI, type of treatment and compliance with the medications), were initially included in the propensity score model.
  6. However, the authors didn’t represent the PS group of patients in the table by the 34 covariates that they mentioned. This would seem to be essential in order for them to make their stated conclusions.
  7. Table 1 shows in non-PS matched patients that patients who were older, female, had more cardiomegaly, arrhythmias, higher cardiovascular risk factors, more comorbidities and other significant disorders had longer SLOS and a higher number of primary ambulatory visits (PAV). This seems intuitive and hardly requires a formal study to understand.
  8. Although the authors mention propensity score-matching, neither Table 1 nor Table 2 shows PS-matched group comparisons. Both the univariate analysis (Table 2) and the multivariable analysis described in the text (no table) were performed on non-PS matched patients.
  9. However, in this study, it would have been more informative if the authors had done the univariate and multivariable analyses in PS-matched groups.
  10. They also should have divided the cohort into derivative subsets and created a “hold-out” sample for validation and then included a separate dataset for external validation. The sample size in this study is relatively large so it there are sufficient patients to do an external validation. 
  11. Also, the authors should differentiate the cardiovascular-related and non-cardiovascular-related healthcare services utilization and the causes of mortality.
  12. I couldn't find any information on the optimal medical therapy for these patients. Rather, the authors simply mentioned this in a superficial, passing manner.However, it would have been informative to know how many patients were on ACE/ACE II inhibitors, beta-blockers, calcium channel blockers, aspirin, and anticoagulant therapy, as well as how many had a stroke and/or atrial fibrillation. Several randomized controlled trials have documented that the method in which these patients are medically managed impacts long-term survival and readmissions.

Reviewer 2 Report

The authors evaluated the association between the extent of healthcare resources utilization (SLOS and PAV) during the first year following AMI and long-term survival. Finally, they expressed that increased SLOS and reduced number of PAV were independently associated with increases risk for mortality. Interestingly, they focused on not only the number of readmissions but a period of readmissions following AMI. This study with big data contained important information for clinical doctors; however, the medical circumstance in Israel cannot completely apply to that in other nations.

The study is overall well-written but we have some specific comments.

Abstract

Line 20: please check the word “SLOS” is proper abbreviation?

Line 20: the sentence seems to be grammatically strange. The better one is “the associations between subsequent hospitalizations and primary ambulatory services utilization within the first year after AMI and long-term mortality” such as the title.

Line 24: you should rephrase “All-cause mortality was set as the primary outcome.

Line 26: you need to insert “,” between (up-to 18 years) and 49.6%.

Line 28: you should unify the words “compared with” or “vs.”

Introduction

Line 36: you need to insert “,” between management and resulting.

Materials and Methods

Line 52-55: How many patients were excluded in each condition 1)-4), respectively?

Line 57: Please provide detailed information of local ethics committee, such as “Negev University Ethics Committee (reference: 20-133)”.

Line 86: why did you determine 10 visits to divide the two groups?

Line 100: The authors set all-cause mortality as the primary outcome. However, they should regard cardiac death as the primary outcome.

Line 102: please provide the detailed info about SPSS, such as (SPSS Inc., Chicago, IL, USA).

Line 103: all of data were under normal distribution?

Line 104: please check typo, percepts→ percent.

Results

Line 121~: the SLOS was not distributed normally, because 5.98±15.5 etc. are strange. The data should be expressed median with the inter-quartile range (IQR). Please evaluate whether all the data are normally distributed or not.

Line 125: the authors repeat the word, “utilization.”

Line 136: I do not understand the reason of “Male patients had lower SLOS with no difference between the sexes in PAV.” Could you describe the topic in Discussion.  

Line 145: the authors combined patients with PCI and CABG. However, I would say that each SLOS is different, thus they should separate PCI and CABG.

Table 1

A large number of patients were included in this study, therefore the result have obtained the significant p values. This means the author has α error. The p values in this table are meaningless?

Reviewer 3 Report

Great paper by Plakht et al. investigating a large number of AMI patients. Interesting topic, and the study has been carried out nicely. 

However, some issues should be adressed:

  1. The patients received 'guideline medical treatment' but according to which guidelines? In most parts of the world, AMI patients receive at least dual antiplatelet therapy. If these patients did not, I'm not sure how the results of this study should be interpreted. 
  2. It is not mentioned in the limitations section, but what do the authors think of the selection bias of including insured patients only and that therapy adherence was validated by prescription use only? Will these factors have influence on the outcomes?
  3. The conclusion should be drawn more hypothetically, since this study has several important limitations. 
  4. If patients are admitted more frequently, what should be done to prevent the higher mortality in these patients? i.e.: what actions should the readers undertake when they become aware of higher hospitalisations/ambulatory visitis? 
